# Evaluating the perceived outcome and impact of an integrated knowledge translation approach in the development of an equity reporting guideline: A cross-sectional survey

Jessica Brown[1], Omar Dewidar[2,3]*, Catherine Chamberlain[4], Luis Gabriel Cuervo[5,6], Holly North Ellingwood[7], Sonya Faber[8], Cindy Feng[9], Damian K. Francis[10], Sarah Funnell[11,12], Elizabeth Ghogomu[13], Billie-Jo Hardy[14,15], Tanya Horsley[16,17], Mwenya Kasonde[18], Michelle Kennedy[19], Tamara Kredo[20], Julian Little[21], Michael Johnson Mahande[22], Zack Marshall[23,24,25], Lawrence Mbuagbaw[26], Miriam Nkangu[27,28,29], Ekwaro A. Obuku[30], Oyekola Oloyede[31], Ebenezer Owusu-Addo[32], Tomás Pantoja[33], Kevin Pottie[34], Anita Rizvi[35], Larissa Shamseer[36,37], Beverley Shea[38], Janice Tufte[39,40], Peter Tugwell[41,42], Zulfiqar Bhutta[43,44], Charles S. Wiysonge[45], Luke Wolfenden[46], Janet Jull[47,48], Vivian Welch[3,49]

1 Faculty of Medicine, University of Ottawa, Ottawa, Ontario, Canada, 2 Temerty Faculty of Medicine, University of Toronto, Toronto, Ontario, Canada, 3 Bruyère Health Research Institute, University of Ottawa, Ontario, Canada, 4 Indigenous Health Equity Unit, Melbourne School of Population and Global Health, The University of Melbourne, Melbourne, Australia, 5 Doctoral Programme on Methodology of Biomedical Research and Public Health, Department of Paediatrics, Obstetrics, Gynaecology and Preventive Medicine and Public Health, Autonomous University of Barcelona, Bellaterra, Cataluña, Spain, 6 National Academy of Medicine of Colombia, Bogotá, Colombia, 7 Department of Law, Department of Psychology, Carleton University, Ottawa, Ontario, Canada, 8 School of Epidemiology and Public Health, University of Ottawa, Ottawa, Ontario, Canada, 9 Community Health and Epidemiology, Dalhousie University, Halifax, Nova Scotia, Canada, 10 School of Health and Human Performance, Center for Health and Social Issues, Georgia College and State University, Milledgeville, Georgia, United States of America, 11 Department of Family Medicine, Queen's University, Kingston, Canada, 12 Department of Family Medicine, Faculty of Medicine, University of Ottawa, Ottawa, Ontario, Canada, 13 Bruyère Health Research Institute, University of Ottawa, Ontario, Canada, 14 Dalla Lana School of Public Health, University of Toronto, Ontario, Canada, 15 Well Living House, Li Ka Shing Knowledge Institute, Toronto, Ontario, Canada, 16 Royal College of Physicians and Surgeons of Canada, Research, Ottawa, Canada, 17 School of Epidemiology and Public Health, Faculty of Medicine, University of Ottawa, Ottawa, Canada, 18 Liverpool School of Tropical Medicine, London, United Kingdom; World Health Organization, Geneva, Switzerland, 19 School of Medicine and Public Health, The University of Newcastle, Callaghan, New South Wales, Australia, 20 Health Systems Research Unit, South African Medical Research Council, Cape Town, South Africa, 21 School of Epidemiology and Public Health, Faculty of Medicine, University of Ottawa, Ottawa, Canada, 22 Department of Epidemiology & Biostatistics, Institute of Public Health, Kilimanjaro Christian Medical College, Moshi, Tanzania, 23 Department of Community Health Sciences, Cumming School of Medicine, University of Calgary, Calgary, Alberta, Canada, 24 Department of Health Research Methods, Evidence and Impact, McMaster University, Hamilton, Ontario, Canada, 25 Department of Anesthesia, McMaster University, Hamilton, Ontario, Canada, 26 Bruyere Health Research Institute, Ottawa, Canada, 27 Africa Centre for Systematic Reviews and Knowledge Translation, College of Health Sciences, Makerere University, Kampala, Uganda, 28 Department of Medicine, School of Medicine, College of Health Sciences, Makerere University, Kampala, Uganda, 29 Faculty of Epidemiology & Population Health, London School of Hygiene & Tropical Medicine, London, United Kingdom, 30 Sefako Makgatho Health Sciences University, Pretoria, South Africa, 31 Institute for Rural Development and Innovation Studies, Kwame Nkrumah University of Science & Technology, Kumasi, Ghana, 32 Department of Family Medicine, Pontificia Universidad Católica de Chile, Santiago, Chile, 33 Department of Family Medicine, Dalhousie University, Halifax, Nova Scotia, Canada, 34 School of Psychology, University of Ottawa, Ottawa, Canada, 35 Knowledge Translation Program, Li Ka Shing Knowledge Institute, St. Michael's Hospital, Toronto, Ontario, Canada, 36 Bruyère Health Research Institute, University of Ottawa, Ottawa, Canada, 37 School of Epidemiology and Public Health, Faculty of Medicine University of Ottawa, Ottawa, Canada, 38 Hassanah Consulting, Seattle, Washington, United States of America, 39 Department of Medicine, and School of Epidemiology and Public Health, Ottawa, Canada, 40 Bruyère Health Research

**Data availability statement:** All available data have been summarized in the manuscript. Data contains potentially identifying or sensitive patient information; therefore, it cannot be shared. For more information, please get in touch with Bruyere Research Ethics Board (REB@bruyere.org).

**Funding:** The STROBE-Equity project (VW) was supported by the Canadian Institutes of Health Research (CIHR), grant number 173269. The funder had no role in the study design, the collection, analysis, and interpretation of data or the writing of the article or the decision to submit it for publication. The first author (JB) was funded by the University of Ottawa through the Medical Student Summer Research Program. VW holds an Applied Public Health Research Chair funded by CIHR-PHAC. The funds partially supported this work. The funder had no role in the design, conduct, or reporting of this study.

**Competing interests:** I have read the journal's policy and the authors of this manuscript have the following competing interests: VW and SF received support from the Canadian Institutes of Health Research grant for the submitted work. Payments were made to Bruyère Health Research Institute. SF serves as a Board Member on College of Family Physicians of Canada and National Circle of Indigenous Medical Education. LGC declares no support from any organization for the submitted work. He has no financial relationships with organizations that might have an interest in the submitted work. He serves as an advisor and editorial board member for journals as a voluntary contribution, with details listed in his CV and ORCID profile. His employer, the Pan American Health Organization (PAHO/WHO), has covered travel incurred as part of his full-time position, and a copyright agreement was submitted through the corresponding author. He holds a retirement fund with BMA (BMP PG) and the United Nations Joint Staff Pension Fund, over which he has no control. His views expressed in this article do not necessarily represent PAHO or WHO policies, and reproduction of the article must avoid implying endorsement

Institute, Ottawa Hospital Research Institute, Ottawa, Canada, **41** University of Ottawa, Ottawa, Canada, **42** Centre for Global Child Health, Hospital for Sick Children, Toronto, Canada, **43** Institute for Global Health & Development, The Aga Khan University, Karachi, Pakistan, **44** Centre for Evidence Based Health Care, Department of Global Health, Stellenbosch University, Cape Town, South Africa, **45** Cochrane South Africa, South African Medical Research Council, Cape Town, South Africa, **46** The University of Newcastle, School of Medicine and Public Health, Newcastle,Australia, **47** Department of Health Sciences, Carleton University, Ottawa, Canada, **48** Faculty of Health Sciences, School of Rehabilitation Therapy, Queen's University, Kingston, Canada, **49** School of Epidemiology and Public Health, Faculty of Medicine University of Ottawa, Ottawa, Canada

* Omar.dewidar@mail.utoronto.ca

## Abstract

Integrated knowledge translation (IKT) involves active engagement of knowledge users in co-producing research, ensuring their perspectives shape study design, analysis, and reporting. This can strengthen justice, equity, diversity, and inclusion (JEDI) considerations. We adopted an IKT approach in developing STROBE-Equity, an equity-focused reporting guidelin extension. The perceived value of embedding JEDI principles in reporting guideline development is unknown. This study examines the team's perceptions on the implementation of the JEDI-enhanced IKT process and its influence on the guideline. We conducted a cross-sectional survey of STROBE-Equity project members (n = 68) between July–August 2024. The 19-item survey assessed disciplinary background, participation, and perceived benefits, challenges, and potential impacts of the JEDI-enhanced IKT approach. Inductive content analysis was used to identify themes, which were quantified with frequencies and percentages. Thirty-one members responded. Most were aged 35–54 (61%), female (55%), based in Canada (35%), and trained in epidemiology (61%). Reported benefits of IKT included integrating diverse perspectives, inclusive representation, and collaborative learning. Challenges involved accessibility and accommodations, consensus-building, and navigating power dynamics between researchers, policymakers, and those with lived experience. Participants perceived that IKT broadened the understanding of social conditions in the development process and facilitated incorporation of end-user perspectives, which they believed would strengthen the credibility and applicability of the guideline. They also noted that this collaborative approach would likely enhance the dissemination and uptake of STROBE-Equity and enhance its acceptability moving forward. A JEDI-enhanced IKT approach was viewed as beneficial to the development of the reporting guideline. Challenges such as accessibility and balancing power dynamics highlight areas where the participatory process could be improved. Future research should continue to refine and evaluate inclusive approaches to guideline development to further advance JEDI in research.

by these organizations. SF is an unpaid Board member of Chacruna Institute for Psychedelic Plant Medicines, and an employee of Angelini Pharma and a partner in the Bioville GmbH. TK is a Cochrane Board member and was an elected trustee since 2020. ZM reports grants from NSERC CREATE grant (Responsible AI), SSHRC Knowledge Synthesis Grants: Gender-Based Violence, and CIHR Catalyst Grant: Moving Upstream: Structural Determinants of Health. KP reports grants from the Public Health Agency of Canada and the Canadian Institutes of Health Research (CIHR) and has been involved in multiple funded research projects related to COVID-19 and health equity, with payments made to Western University. He received consulting fees from CSL Behring for Ferinject clinical consultancy, honoraria for a guest lecture in Community Diabetes CME, and holds a pending patent related to health information-based communities and knowledge incentive systems. He also holds leadership roles in The Canadian Collaboration for Immigrant and Refugee Health, Cochrane Canada, and MSF Canada, all of which are unpaid. All authors have completed the ICMJE uniform disclosure form at www.icmje.org/disclosure-of-interest. The remaining authors have nothing to declare.

**Abbreviations:** IKT, Integrated Knowledge Translation; STROBE, Strengthening the reporting of observational studies in epidemiology

## Introduction

Addressing health equity is a global priority, as emphasized in the United Nations Sustainable Development Goals (SDGs) and operationalized in the mandates of UN agencies such as the World Health Organization (WHO). In 2015, 193 nations committed to the SDGs, outlining a 2030 agenda for sustainable development [1]. In Canada, this global commitment is echoed within national priorities, including those of its primary funding agency for health-related research, the Canadian Institutes of Health Research (CIHR) [2].

Despite growing recognition, advancing health equity remains a persistent challenge. Progress requires conceptual clarity in the rationale for equity assessments, adherence to methodological standards, and complete and consistent reporting of data. However, the evidence base is undermined by insufficient and inconsistent reporting of equity-related data, limiting user's ability to effectively evaluate and address inequities. Equity-related assessments are further hindered by recent socio-political developments, including the dismantling of equity, diversity and inclusion (EDI) initiatives and the rollback equity-focused research infrastructure in the USA. Simultaneously, the defunding of global development assistance aimed at reducing health inequities has impacted global efforts.

Health inequities are differences in the opportunities groups have to achieve optimal health, leading to unfair and avoidable differences in health outcomes [3–5]. They are driven by the injustices that exist systemically and structurally in systems, which are associated to an unbalanced distribution of power and privilege [4]. Factors which are associated with inequities in opportunities for health have been described by the acronym PROGRESS-Plus; short for Place of residence, Race/ethnicity/culture/language, Occupation, Gender or sex, Religion, Education, Socioeconomic status and Social capital [6]. These factors interact, and multiple disadvantages may compound the experience of inequities. Health inequities are driven by social processes such as colonization, racism, sexism and ableism which act to prevent more equitable distribution of resources and further marginalize already disenfranchised people and communities, compounding their socioeconomic disadvantage and entrenching health disparities further [7]. These inequities can be exacerbated by research that lacks a central focus on inclusivity and equity (e.g., selection criteria and recruitment of participants), resulting in limited uptake and less research to inform health decisions.

'Knowledge users' are individuals or organizations involved in the research process whose experience is directly connected to the research topic, and may be impacted by the research itself. Knowledge users may be involved in the entire study process from conception of the study question, extending to dissemination and implementation of the research findings or recommendations [8]. Integrated knowledge translation (IKT) offers an approach to advance health equity by actively engaging diverse knowledge users in research co-production [8,9]. IKT has been shown to enhance the usefulness and relevance of research [10]. This approach seeks to support lasting change by actively engaging with the perspectives and knowledge that people bring to research [8,9]. However, challenges such as

difficulties incorporating diverse perspectives and uncertainties about the overall impact of IKT on research outcomes, remain substantive concerns [8].

Observational studies are the most common type of health-related research design. These studies investigate the occurrence of health-related events within a specified population and time-period without researchers controlling the exposure(s) or intervention [11]. These studies are particularly beneficial to inform health equity considerations as they routinely collect data related to systemic or structural inequities. However, reporting practices for such studies are inconsistent, limiting the use of the data for advancing equity-oriented research and policy [4].

To address the inconsistent reporting of equity-relevant data in observational studies [12–14], our team developed an equity extension for the STROBE (STrengthening the Reporting of OBservational studies in Epidemiology) guideline. We accomplished this by using an IKT approach, with knowledge users as part of an interdisciplinary and international research team, and adopted and integrated principles of justice, equity, diversity and inclusion (JEDI) throughout the reporting guideline development process. Because our focus was to improve health equity reporting, we also explicitly sought diversity in our team across career stages, gender, geographies, lived experience of health inequities and disciplinary perspectives.

The aim of this study was to evaluate the perceptions of academics and knowledge user members of the STROBE-Equity research team on the benefits, challenges and potential impact of the JEDI-enhanced IKT approach applied in the development of the guideline.

## Methods

### Study design

The STROBE-Equity project involves two parallel streams, one to develop a global guideline and another to develop an Indigenous-specific guideline. The protocol for developing the STROBE-Equity extension is published elsewhere [4]. In brief, the project involved four phases: 1) empirical studies assessing the reporting of health equity in published observational studies [2,14–16] seeking wide international feedback on items by carrying out a global online survey (in review), [3] establishing consensus amongst knowledge users and researchers (in review), and [4] widely disseminating the equity extension and seeking endorsement from relevant knowledge users. We ran a parallel process for Indigenous health research and health equity reporting, led by an Indigenous Steering Committee. The conceptualization and development of the project is visualized in Fig 1. The following study is reported according to the Consensus-Based Checklist for Reporting of Survey Studies (CROSS) reporting guideline [17] (S1 Table A in S1 Text).

### Survey development

In consultation with our executive team members (JJ, VW, LM, SF), we developed a 19-question online survey using the SurveyMonkey tool (https://www.surveymonkey.com/) to evaluate the research team's experiences with the IKT approach and suggestions for improvement. The survey was adapted from a previous survey deployed for evaluation of IKT in the context of a similar reporting guideline on equity in randomized trials; CONSolidated reporting Of Randomized Trials (CONSORT)-Equity [18]. Survey questions were designed to elicit respondents' understandings of their experiences with, and perceptions of, the JEDI-enhanced IKT process including its percieved benefits, challenges and potential impact. It included four Likert-scale questions assessing the extent of engagement in the IKT process, with response options ranging from 1 (no engagement) to 5 (full engagement). Additionally, the survey contained nine items capturing respondents' disciplinary and sociodemographic characteristics, as well as six open-ended questions exploring perceived benefits, challenges, and the overall impact of the IKT process, including the emphasis on reflecting JEDI within the team. We did not conduct reliability assessments of the survey items, as the survey primarily collected nominal data (e.g., discipline, gender, location) and did not include continuous or scale-based items for which such

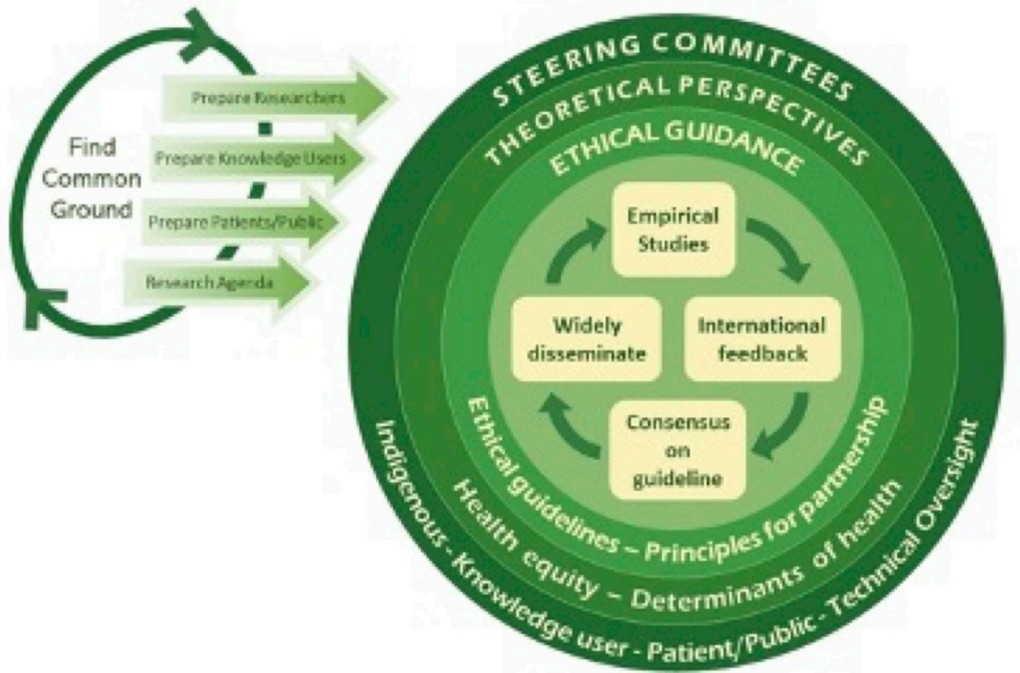

**Fig 1. Inclusive governance plan of the STROBE-Equity project.**

tests would be appropriate. The sociodemographic questions were optional and included questions about the country of birth, country where currently living, ethnicity, age, gender identity, and occupation. The survey questions were reviewed by the members of the executive team, and pilot tested among them before deployment and improvements regarding survey clarity were made iteratively.

## Data collection methods

The STROBE-Equity team is an international collaboration consisting of members from eight different countries. Members of the team were intentionally selected to ensure that individuals who lived with chronic health conditions (public partners), researchers across diverse disciplines and decision-makers (e.g., journal editors, funders, policy decision-makers) were represented [19]. We also explicitly sought diversity across disciplinary backgrounds, gender, ethnicity, country of origin, career stage, and types of knowledge user perspectives including those with lived experience of inequities. Our team included members of the public, authors of observational studies, statisticians, social scientists, epidemiologists, methodologists, funders, healthcare practitioners, and ethicists. The survey questionnaire is described in S1 Text A S1 Text.

## Survey administration

All 68 STROBE-equity team members, including members of all steering committees, were asked to anonymously participate in the survey following the development of the STROBE-Equity reporting guideline. We sent the survey on July 17th, 2024, and closed it on August 26th, 2024, after three reminders.

## Governance of the IKT process

We developed an executive team consisting of the four principal investigators and all trainees on the project which met monthly. We formed four steering committees including a technical oversight committee, knowledge users committee,

patient and public committee and the Indigenous steering committee which met quarterly. We decided to have a separate committee for patients and the public to enhance relevance of discussions, promote their engagement, and give more weight to their voices. The members of these four committees had diverse backgrounds and included epidemiologists, methodologists, journal editors, policymakers, health professionals, from public and private sectors, representing different geographic regions and World Bank income levels. The Indigenous steering committee included Indigenous scholars from Australia, Canada, and New Zealand included an Indigenous Elder from Canada and ally researchers.

We developed a governance plan and list of planned projects, mapping out the activities, roles, levels of engagement, and timelines for the global and Indigenous streams of research. Terms of reference were drawn up for an authorship and publication strategy. All team members were invited to be authors, and we followed the International Committee of Medical Journal Editors (ICMJE) guidelines for authorship [20]. We prepared a list of possible publications led by trainees and the team members were invited to sign up for those in which they were interested in participating as authors.

At a virtual launch meeting (held at two times to accommodate global time zones), all members agreed on the terms of reference, the activities, roles, and timelines. It was agreed that the global and Indigenous streams would run separately thus allowing each the freedom to choose approaches to meet their goals, while contributing to processes that support good data governance, as only members of the group were included in the research processes. We developed a logo representing the two streams which was designed by an Indigenous artist (Claire Brascoupé, S1 Table B S1 Text). Minutes from each meeting were circulated to committee members and a quarterly newsletter was circulated to inform all members and to cross-pollinate ideas from the global and the Indigenous research streams and different steering committees.

We applied the Collaborative Research Framework to guide our the evaluation of our IKT approach [21]. This framework accommodated formal and structured partnering between knowledge users and researchers in the preparation, planning, and conduct stages of the empirical studies informing the STROBE-Equity reporting guideline. The framework assumes that IKT is facilitated when team members are prepared for ongoing, iterative knowledge exchange and shared learning. Consistent with this, our approach involved two core components: 1) establishing guiding features for co-production through an equity-driven governance structure, and 2) defining and operationalizing research activites that enabled joint production of relevant, usable and equitable evidence. Our selection for this framework was based on previous experience with the framework to guide reporting guideline-focused research involving diverse knowledge users and we utilized it as the conceptual foundation for our IKT approach, as cited in our reporting guideline protocol [4,22].

## Patient and public involvement

Patient engagement was reported according to GRIPP2 checklist [23] (Table B). Our Patient and Public Steering Committee was co-led by three team members who had been engaged since the funding acquisition stage and had lived experience of health inequities. They also have experience conducting health equity research. Members were compensated annually throughout the length of the project and regularly engaged through quarterly meetings and project newsletters. They contributed to survey development, interpretation of findings, and refinement of language to reflect lived experience. Their input throughout the grant application, team orientation and project design meetings aimed to bring experiences of patients and public members into the design of the research as well as the interpretation of findings.

## Statistical analysis

Descriptive statistics (frequencies and percentages) were used to summarize responses to close-ended Likert-scale questions. For each analysis, calculations were performed using all available responses for the specific variables, rather than excluding cases with any missing data. Open ended questions were analyzed using inductive content analysis [24,25]. This process involves pooling responses for each of the six open-ended questions (questions 7–12 in the survey, S1 Text A S1 Text), then developing codes within each question. We developed codes to describe the responses within each of these categories and reported the frequency counts of the type of responses with illustrative quotes. One researcher (OD)

conducted inductive coding for each open-ended question. A second researcher (JB) independently reviewed the codes. Discrepancies were discussed and resolved through consensus.

## Results

### Characteristics of survey respondents

The survey was distributed among STROBE-Equity members (n = 68) from July 17 to August 26, 2024. Of 68 team members, thirty-one people responded to the survey as reported in Table 1 (response rate = 46%). The percentages in the table are reflective of that total. Half of respondents attended 50–75% of STROBE project meetings, and while some were engaged in both the global and Indigenous streams, most had only participated in the former. Most respondents (84%) felt that they were "engaged" in the STROBE project, defined as a score of ≥3 on a 5-point engagement scale (with 5 indicating "totally engaged"). Most (87%) reported being "satisfied" or "very satisfied" or "totally satisfied" with their level of engagement on the project. The majority identified as being principal investigators and members of the research team (84%), or those that advocated for health research. Others represented smaller proportions of the group, including peer reviewed journal editors (23%), patients (16%), and policymakers (16%). In terms of educational background, clinical epidemiology was the most common (19 respondents, 61%), complemented by expertise from clinicians, public health specialists, and policy advocates.

**Inductive content analysis.** We analyzed 107 open-ended comments from 31 respondents across the six open-ended questions. These questions asked about the benefits, challenges, and impact of the IKT process used to reflect JEDI within the team and motivations for participation. Themes were identified from the open-ended questions in four categories related to benefits, challenges, impact and motivations (Table 2).

### Diverse and comprehensive perspectives

In the design of the reporting guideline, respondents found it important to include diverse and comprehensive perspectives. This was reported by most individuals (n = 20, 65%); *"diverse perspectives are vital for accessible, acceptable guidelines"*. Others mentioned words such as *respect, richness, acceptability,* and *avoid blind spots or misinterpretation of results*.

### Engage underrepresented groups in research

Engagement of underrepresented groups and inclusive representation throughout the research process was appreciated by (n = 12) 39% of the respondents. Respondents shared that the process gave them the opportunity to hear distinct voices. The participatory approach allowed for further engagement of "*populations who have not been represented*" or historically excluded from these discussions. Moreover, hearing from those with lived experience was seen as "*eye opening*" and helped incorporate discussion from people with lived experience of health inequities. Respondents reported that it amplified their experience, addressed the needs of interest holders involved in dissemination of the IKT process, and identified the social determinants of health involved in decision making processes.

### Collaborative learning experience

Of the 31 respondents, (n = 6) 19% shared that by being involved, it provided a collaborative learning environment allowing the team to, as reported, *"learn from each other"*. In addition, participation in the project challenged the thoughts and biases about individuals and amongst each other. It was appreciated that there was complete transparency with how each decision was made and how the team processes facilitated interpersonal relationship building. Overall, it extended the thinking of both the team and the individual.

**Perceived challenges of IKT.** Themes related to challenges of IKT were analyzed from 41 open-ended comments from 31 respondents.

**Table 1.** Characteristics of survey respondents (N=31).

| Respondent Characteristics | n (%) |
|---|---|
| **Meetings attended** | |
| 0-25% | 4 (13) |
| 25-50% | 7 (23) |
| 50-75% | 17 (55) |
| 75%+ | 3 (10) |
| **Engagement Level** | |
| Somewhat | 6 (19) |
| Engaged | 17 (55) |
| Very | 7 (23) |
| Totally | 1 (3) |
| **Current Country** | |
| Australia | 1 (3) |
| Canada | 11 (35) |
| Germany | 1 (3) |
| Ireland | 1 (3) |
| Lesotho | 1 (3) |
| New Zealand | 2 (7) |
| South Africa | 2 (7) |
| Switzerland | 1 (3) |
| Tanzania | 1 (3) |
| Uganda | 1 (3) |
| United Kingdom | 2 (7) |
| United States | 4 (13) |
| Not reported | 3 (10) |
| **Country of birth** | |
| Australia | 2 (7) |
| Canada | 8 (27) |
| China | 1 (3) |
| Germany | 1 (3) |
| New Zealand | 1 (3) |
| United Kingdom | 3 (10) |
| United States | 6 (20) |
| Other (includes Tanzania, Jamaica, S. Africa, Pakistan, Egypt, Uganda, Nigeria, Lebanon) | 8 (27) |
| Not reported | 1 (3) |
| **Ethnic origin** | |
| Western Europe | 15 (50) |
| Eastern Europe | 2 (6) |
| Sub-Saharan Africa | 6 (20) |
| Southeast Asia | 2 (6) |
| Oceania | 4 (13) |
| North America | 4 (13) |
| Caribbean | 2 (6) |
| Other | 4 (13) |
| **Ethnicity** | |
| Asian | 1 (3) |

*(Continued)*

**Table 1.** (Continued)

| Respondent Characteristics | n (%) |
| --- | --- |
| Black | 7 (23) |
| Indigenous | 4 (13) |
| Middle Eastern | 1 (3) |
| White | 14 (47) |
| No disclosure | 1 (3) |
| Other | 4 (13) |
| **Age** | |
| 18-34 | 1 (3) |
| 35-54 | 19 (61) |
| 55-64 | 4 (13) |
| 65-74 | 5 (17) |
| 75-84 | 1 (3) |
| **Gender identity** | |
| Male | 13 (42) |
| Transgender | 1 (3) |
| Female | 17 (55) |
| **Occupation** | |
| Academics | 1 (35) |
| Pharmacy | 1 (3) |
| Physician | 2 (6) |
| Public Health | 4 (13) |
| Research | 9 (29) |
| Self-Employed | 1 (3) |
| **Group** | |
| Patients | 5 (16) |
| Payers of services | 2 (6) |
| Journal Editors | 7 (23) |
| Policymakers | 5 (16) |
| Product makers | 1 (3) |
| Program managers | 1 (3) |
| Providers | 3 (10) |
| Public | 5 (16) |
| Researchers | 26 (84) |
| **Background** | |
| Statistics | 2 (6) |
| Epidemiology | 22 (71) |
| Law | 1 (3) |
| Clinician | 10 (32) |
| Public Health | 3 (10) |
| Other | 8 (26) |

**Note:** Percentages for categories such as "Occupation", "Group" and "Background" may total to more than 100 since the categories are not mutually exclusive.

**Table 2.** Summary of open-ended survey responses on perceived benefits, challenges, and impacts of the justice, equity, diversity and inclusion enhanced IKT approach in STROBE-Equity development (N = 31).

| Perceptions | Themes |
|---|---|
| Perceived benefits of using a participatory approach to reflect justice, equity, diversity and inclusion in developing the STROBE-Equity extension? | 1. Diverse and comprehensive perspectives in the design of the guideline (20/31 = 65%)<br>2. Engage underrepresented groups in research/ inclusive representation (12/31 = 39%)<br>3. Collaborative learning experience (6/31 = 19%) |
| Perceived challenges of using a participatory approach to reflect justice, equity, diversity and inclusion in developing the STROBE-Equity extension? | 1. Challenges of enabling global participation (9/31 = 29%)<br>2. Barriers to Consensus building (8/31 = 26%)<br>3. Differences in power dynamics (6/31 = 19%) |
| Perceived impact of using a participatory approach to reflect justice, equity, diversity and inclusion in developing the STROBE-Equity extension | 1. Increase the opportunities for dissemination and uptake (12/31 = 39%)<br>2. Advance the decolonization of views and perspectives (14/31 = 45%)<br>3. Improve credibility and comprehensiveness of guideline (17/31 = 55%) |
| Motivations for participating in STROBE | 1. Promoting equity (14/31 = 45%)<br>2. Learning (education/team dynamics) (10/31 = 32%)<br>3. Advancing research agendas and methodologies (11/31 = 35%) |

## Challenges with enabling participation of a global team

About one third of respondents mentioned a significant challenge related to promoting global participation such as the logistics of coordinating meetings and enabling the wider team to participate (n = 9, 29%). Concerns such as having access to equipment or internet were shared. Also, differences in time zones were a factor with the team being dispersed globally. Other comments mentioned the concept of time allowance, and how having more time to work on the project would have promoted more space to "*think deeply*" about approaches and solutions. Project costs were also a challenge shared by respondents. project costs were identified as a limiting factor to inclusive participation. Respondents emphasized that costs related to internet access, training, and equitable support for diverse contributors posed tangible obstacles to engagement. One respondent noted that there are "*several challenges, such as training for certain members, time allowance for certain members, consensus building, [and] cost to support interest holders.*" Another categorized the challenge as one of "*accommodating everyone (time zones, access to internet, cost),*" emphasizing the additional financial and logistical barriers. A few respondents also suggested that in-person meetings could have "*heightened engagement*" and allowed for more in-depth discussions, although this was acknowledged as potentially exclusionary due to associated travel and participation costs.

## Barriers to consensus building

Some respondents (n = 8, 26%) found difficulty in contributing to and agreeing on group conclusions and truly co-developing the work at times. For example, some found that "*previous conflict amongst other members hindered full participation in some meetings*". Others pointed out the difficulties that may come into play when working with a diverse team. For example, it was mentioned that there was a predominance of Canadians on the team, and asked "*on what factors were diversity sought?*". Additionally, some team members felt that differences in educational backgrounds and varying levels of familiarity with technical content limited their ability to fully engage in discussions or contribute meaningfully to project development. One respondent observed that the group faced "*several challenges, such as training for certain members*", suggesting that varying technical capacities and limited access to training infrastructure may have

affected equitable participation. Another emphasized that barriers such as "access to equipment/internet" and time zone constraints added further complexity to participation, particularly for contributors in settings with limited resources.

### Power dynamics

By working with a global, diverse team, it prompted us to reconsider how to create more inclusive spaces for all perspectives. A few respondents (n = 6, 19%) said that at times, because of the varying levels of lived experiences, *"changes were met with resistance"* since it was hard at times for those who do not directly experience discrimination to relate to what was being requested. Another concern was that many authors are from high-income countries, and the potential bias that came from not necessarily knowing if the right representatives were being included from the diverse groups in which we seek to engage. Perhaps the most intriguing comment suggested a potential cause of resistance from the public, in which by including an equity extension there remains a *"fear of those in power that diversity is a zero-sum game",* as identified by a survey participant.

**Perceived impact of using a participatory approach to reflect justice, equity, diversity and inclusion enhanced IKT.** These themes are based on analysis of 23 open-ended comments from 31 people.

### Increases the opportunities for dissemination and uptake

A number of respondents (n = 12, 39%) reported that the extension would likely increase opportunities for dissemination and most importantly that "*a participatory approach helps ensure the end product will be used*". This reflected the belief that the collaborative nature of the IKT would promote buy-in. Key words used were *validity*, *generalizability* and *relevance*. Others mentioned that the end product was of quality, held meaning, and was structured in its development. These were factors they believed could enhance future acceptance and application.

### Advance the decolonization of views and perspectives

Reported by (n = 14) 45% of the team was the ability of this project to advance the diversification of perspectives, allowing for better representation of the populations we seek to serve, addressing needs and giving people a voice. Also, the JEDI-enhanced IKT process promotes application and integration of equity concepts, diversity in ideas, and provides opportunities to advance team and individual thinking [26]. In addition, the diversity of the team empowered multidisciplinary learning and idea sharing, as experts gathered from low to high income economies to discuss potential ideas during project development.

### Improved credibility and comprehensiveness of guideline

The concept of applicability was also emphasized, with improved credibility and comprehensibility of the guidelines. This was reported by most respondents (n = 17, 55%). Using a participatory approach encouraged its usefulness in real life applications and promoted a sense of collective ownership. Comments also highlighted how the participatory approach considered different levels of governance and how to adopt shared governance. One respondent wrote that it *"addresses conflicts between stakeholders… [and] stresses to stakeholders their strengths and achievements".*

**Motivations.** When asked about motivations for and perceived benefits from participating in the STROBE Equity project, individuals reported that they were able to see the promotion of health equity at the center of research, supporting the equity agenda and influencing dialogue surrounding equity. Respondents commented on how the IKT approach brought in Indigenous voices and allowed those with lived experience to share their story. Respondents also appreciated how it helped with their learning and team dynamics, including their own education, gaining different perspectives, and elevating their own thinking.

Participation was also viewed as directly relevant to advancing the research agenda and improving methodological quality. One respondent highlighted that the initiative involved *"a structured process… employing empirical survey and online survey,"* which helped achieve improved internal validity and strengthened the rigor of the guideline. Others saw their involvement as an opportunity to contribute to developing methodological standards that shape how equity is

understood and reported in observational studies. One respondent noted that *"guidelines are one of the most effective ways as a researcher that one can influence how research generally is done and reported; and influence the collection and reporting of equity-relevant evidence."* Another respondent reported how engaging in this process helped their future career.

**Changes to the IKT process.** An additional survey question asked if respondents would have changed anything about the participatory approach. Similar to the challenges experienced, issues with accessibility and online meetings were mentioned by 29% of respondents. Positionality was also of note. Although only 6% shared this thought, it was felt that expert contributors could have been tapped to contribute more substantially towards the writing process. Not many respondents had generalized comments about the IKT process, but some mentioned their appreciation for the team dynamics and leadership, sharing "*they were incredible people to work with*". Another comment was that the participatory approach produced a quality end-product.

## Discussion

Using a JEDI-enhanced IKT approach, our international team collaboratively developed the STROBE Equity reporting guidelines which intend to improve the reporting of health equity in observational studies. Herin, we provide insights into how this approach was percieved by members of our international team. Participants perceived the JEDI-enhanced IKT process as a useful strategy for integrating diverse perspectives and informing the development of the guideline. They also also identified several challenges when adopting this approach such as accessibility, difficulty accommodating all members of the team, and navigating power dynamics, which highlight areas for improvement in future participatory research.

STROBE Equity team members appreciated how the IKT facilitated the inclusion of a wide range a range of perspectives in the development process. Respondents described how their engagement broadened their thinking by exposing them to diverse perspectives on social issues. Several participants noted that the IKT approach enabled meaningful involvement of groups that are often underrepresented in research, creating a space for voices that are frequently overlooked. Others appreciated the collaborative learning experiences the process brought to the team. By learning through others, the IKT approach strengthened relationship building and created a strong team dynamic. The importance of relationship building is mentioned in other studies, underlining how IKT fostered partnerships for future collaborations and overcoming differences to deepen their understanding of shared areas of interest [27]. It is of note that these relationships can last for years, creating a network and an invaluable resource for knowledge-sharing [28]. Other studies also mentioned that researchers involved in an IKT approach benefitted from learning about the context of the knowledge users [29].

There were some challenges posed by respondents in reference to the diversity of the team as well as the incorporation of knowledge users in the project. Difficulties with enabling global participation were reported, including time zone differences for meetings or access to electronic equipment. For example, many respondents noted that in-person meetings would have been highly beneficial; however, this was not feasible in view of cost constraints and involvement of numerous global partners. Respondents also shared challenges with consensus building, at times finding it hard to reach group consensus. This included some team members describing they lacked the epidemiological knowledge to fully participate in discussions and reach a conclusion. Lastly, unconscious bias was perceived to influence some aspects of team dynamics and decision-making. For example, respondents noted that when individuals do not have lived experience of discrimination or marginalization, they may find it challenging to recognize or fully appreciate equity-related concerns. As one respondent reflected, *"because it is difficult for those who do not experience discrimination or bias to see and assess it, some of the suggested changes were met with resistance."* This suggests that unacknowledged privilege and limited awareness of inequities may have contributed to resistance when equity-promoting changes were proposed, highlighting the importance of reflexivity and representational diversity in collaborative research settings. Similar IKT studies found

challenges with coordination and logistics, team relationships and shared understanding, and external constraints such as changes to political or personal priorities [27]. For example, it was found that conflicting priorities or identifying the right key partners are some of the most deleterious factors to the success of IKT projects. Future researchers wishing to engage in IKT need to be aware of these challenges [27].

The global push toward equity-oriented research is now facing renewed political headwinds. Political shifts—such as the U.S. administration's planned rollback of justice, equity, diversity, and inclusion initiatives in 2025—pose a significant threat to participatory research approaches like IKT, which depend on the engagement of diverse interest holders to promote equitable knowledge production and application [30,31]. Dismantling these equity initiatives will undermine the inclusion of marginalized voices in research, perpetuating historical inequities and diminishing the relevance of research to real-world, diverse populations [3]. Considering these challenges, future IKT efforts must prioritize strategies that insulate participatory structures from political interference—such as embedding equity frameworks within independent research governance models and securing resilient funding sources.

The STROBE Equity project and its participatory approach were perceived to have an impact on future uptake and research potential. Most importantly, many respondents stated that an IKT approach allows for increased opportunity for dissemination and uptake opportunities and end product use. By including end users in the project development, relevance and validity are strengthened. It also promotes the diversification of perspectives, representing populations from all levels of the economy. Giving a voice to unique voices was reported as significant. Lastly, most reported that IKT improved the credibility of the guidelines. This inclusive approach promoted its usefulness in real life contexts. These findings are similar to other studies which have shown that IKT contributes to a sense of community "ownership" of the issues at play and using action to promote social change. It creates better quality research outputs and builds on the capacity of the interest holder groups involved [27,32,33]. Having early involvement of patients and public from the conceptualization of the project gave ample opportunity to maximize equity considerations with the project [34]. Furthermore, our emphasis on membership diversity contributed to broad perspectives that have enriched the development of the Equity extension and the results of this study. We encourage future researchers to include individuals with lived experiences of inequities and other knowledge users throughout their research process, and to reflect on the strengths, challenges and impact that this engagement brings.

## Strengths and limitations

We used a pre-tested survey to elicit opinions from our STROBE-equity research team and steering committee members. Three reminders were sent to team members to improve response rates. By involving the broader team in interpreting the results, we were able to achieve enhance credibility and richness of the inductive content analysis, promoting a comprehensive understanding of the themes. Our response rate of 46% is a limitation. However, the representation of academic and knowledge users team members and geographic diversity was reflective of our overall team and we did not detect a disproportionate lack of participation by geographical location. Nonetheless, the small number of LMIC respondents limits the extent to which the survey captured perspectives from LMICs. Future studies could be conducted to investigate reasons for non-response and suggest potential solutions. While the online survey was limited in its ability to provide in-depth information compared to interview studies, it proved an efficient and appropriate method for gathering perceptions from a large, geographically dispersed, and diverse international team, aligning with our aim to broadly evaluate the IKT process [27].

## Conclusions

We found that the respondents felt that the IKT process was beneficial to the research process and increased relevance of the end-product. Of note, most respondents shared that by involving knowledge users and those directly impacted by the product of the work, community uptake is promoted, and study validity and generalizability

is strengthened. The challenges of an IKT approach to reflect justice, equity, diversity and inclusion illustrate the need to develop and continuously monitor and evaluate creative strategies to enable equitable knowledge co-production.

**Contributions to the literature**

- Integrated knowledge translation (IKT) involves the active engagement of knowledge users in the co-production of research, to enhance the usefulness and relevance of the research that they are meant to use.

- We describe the potential benefits, challenges and percieved impact of using an IKT approach in developing an evidence-based equity extension of the STROBE (STrengthening the Reporting of OBservational studies in Epidemiol-ogy) reporting guideline.

- Our findings can contribute to establishing best practices of engaging and involving knowledge users in research.

**Supporting information**

**S1 Text. S1 Table A**. Checklist for reporting of survey studies (CROSS). **S1 Text A.** Survey Questions. **S1 Table B.** GRIPP2 reporting checklist.
(DOCX)

**Author contributions**

**Conceptualization:** Jessica Brown, Holly North Ellingwood, Vivian Welch.

**Data curation:** Jessica Brown.

**Formal analysis:** Omar Dewidar.

**Funding acquisition:** Vivian Welch.

**Investigation:** Jessica Brown, Omar Dewidar.

**Methodology:** Jessica Brown, Omar Dewidar, Catherine Chamberlain, Luis Gabriel Cuervo, Holly North Ellingwood, Sonya Faber, Damian K Francis, Sarah Funnell, Cindy Feng, Elizabeth Ghogomu, Billie-Jo Hardy, Tanya Horsley, Mwenya Kasonde, Michelle Kennedy, Tamara Kredo, Julian Little, Michael Johnson Mahande, Zach Marshall, Lawrence Mbuagbaw, Miriam Nkangu, Ekwaro A. Obuku, Oyekola Oloyede, Ebenezer Owusu-Addo, Tomás Pantoja, Kevin Pottie, Anita Rizvi, Larissa Shamseer, Beverley Shea, Janice Tufte, Peter Tugwell, Zulfiqar Bhutta, Charles S Wiysonge, Luke Wolfenden, Janet Jull.

**Supervision:** Vivian Welch.

**Validation:** Omar Dewidar.

**Visualization:** Jessica Brown.

**Writing – original draft:** Jessica Brown.

**Writing – review & editing:** Omar Dewidar, Catherine Chamberlain, Luis Gabriel Cuervo, Holly North Ellingwood, Sonya Faber, Damian K Francis, Sarah Funnell, Cindy Feng, Elizabeth Ghogomu, Billie-Jo Hardy, Tanya Horsley, Mwenya Kasonde, Michelle Kennedy, Tamara Kredo, Julian Little, Michael Johnson Mahande, Zach Marshall, Lawrence Mbuagbaw, Miriam Nkangu, Ekwaro A. Obuku, Oyekola Oloyede, Ebenezer Owusu-Addo, Tomás Pantoja, Kevin Pottie, Anita Rizvi, Larissa Shamseer, Beverley Shea, Janice Tufte, Peter Tugwell, Zulfiqar Bhutta, Charles S Wiysonge, Luke Wolfenden, Janet Jull, Vivian Welch.

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
