## [Decision Letter · Decision Letter 0]

5 Nov 2025

PGPH-D-25-02783

Evaluating the outcome and impact of an integrated knowledge translation approach in the development of an equity reporting guideline: a cross-sectional survey

Dear Dr. Dewidar,

Thank you for submitting your manuscript to PLOS Global Public Health. After careful consideration, we feel that it has merit but does not fully meet PLOS Global Public Health’s publication criteria as it currently stands. Therefore, we invite you to submit a revised version of the manuscript that addresses the points raised during the review process.

Please review the reviewers' comments and revise your manuscript to address them, providing a point-by-point response to the reviewers upon resubmission.

We look forward to receiving your revised manuscript.

Kind regards,

Sarah Jose, Ph.D.

Staff Editor

Journal Requirements:

1. Please provide additional details regarding participant consent. In the ethics statement in the Methods and online submission information, please ensure that you have specified (1) whether consent was informed and (2) what type you obtained (for instance, written or verbal, and if verbal, how it was documented and witnessed). If your study included minors, state whether you obtained consent from parents or guardians. If the need for consent was waived by the ethics committee, please include this information.

2. Please provide a detailed online Financial Disclosure statement. This is published with the article. It must therefore be completed in full sentences and contain the exact wording you wish to be published.

a) State the initials, alongside each funding source, of each author to receive each grant. For example: “This work was supported by the National Institutes of Health (####### to AM; ###### to CJ) and the National Science Foundation (###### to AM).”

For more information, please go to our submission guidelines:

https://journals.plos.org/globalpublichealth/s/submission-guidelines#loc-financial-disclosure-statement

3. Please ensure that the funders and grant numbers match between the Financial Disclosure field and the Funding Information tab in your submission form. Note that the funders must be provided in the same order in both places as well.

4. We note that you have indicated that there are restrictions to data sharing for this study. For studies involving human research participant data or other sensitive data, we encourage authors to share de-identified or anonymized data. However, when data cannot be publicly shared for ethical reasons, we allow authors to make their data sets available upon request. For information on unacceptable data access restrictions, please see http://journals.plos.org/globalpublichealth/s/data-availability#loc-unacceptable-data-access-restrictions.

b) If there are no restrictions, please upload the minimal anonymized data set necessary to replicate your study findings to a stable, public repository and provide us with the relevant URLs, DOIs, or accession numbers. Please see http://www.bmj.com/content/340/bmj.c181.long for guidelines on how to de-identify and prepare clinical data for publication. For a list of recommended repositories, please see https://journals.plos.org/globalpublichealth/s/recommended-repositories. You also have the option of uploading the data as Supporting Information files, but we would recommend depositing data directly to a data repository if possible.

5. Please provide separate figure files in .tif or .eps format only and remove any figures embedded in your manuscript file. Please also ensure that all files are under our size limit of 10MB. Please leave the figure captions in the manuscript.

6. We notice that your supplementary materials are included in the manuscript file. Please remove them and upload them with the file type 'Supporting Information'. Please ensure that each Supporting Information file has a legend listed in the manuscript before or after the references list.

7. Some material included in your submission may be copyrighted. According to PLOS’s copyright policy, authors who use figures or other material (e.g., graphics, clipart, maps) from another author or copyright holder must demonstrate or obtain permission to publish this material under the Creative Commons Attribution 4.0 International (CC BY 4.0) License used by PLOS journals. Please closely review the details of PLOS’s copyright requirements here: PLOS Licenses and Copyright. If you need to request permissions from a copyright holder, you may use PLOS's Copyright Content Permission form.

Potential Copyright Issues:

Figure 2: please (a) provide a direct link to the base layer of the map (i.e., the country or region border shape) and ensure this is also included in the figure legend; and (b) provide a link to the terms of use / license information for the base layer image or shapefile. We cannot publish proprietary or copyrighted maps (e.g. Google Maps, Mapquest) and the terms of use for your map base layer must be compatible with our CC-BY 4.0 license.

Additional Editor Comments (if provided):

Reviewers' comments:

Reviewer's Responses to Questions

**Comments to the Author**

1. Does this manuscript meet PLOS Global Public Health’s publication criteria ? Is the manuscript technically sound, and do the data support the conclusions? The manuscript must describe methodologically and ethically rigorous research with conclusions that are appropriately drawn based on the data presented.

Reviewer #1: Yes

Reviewer #2: Partly

2. Has the statistical analysis been performed appropriately and rigorously?

Reviewer #1: N/A

Reviewer #2: Yes

3. Have the authors made all data underlying the findings in their manuscript fully available (please refer to the Data Availability Statement at the start of the manuscript PDF file)?

Reviewer #1: No

Reviewer #2: Yes

4. Is the manuscript presented in an intelligible fashion and written in standard English?

Reviewer #1: Yes

Reviewer #2: Yes

5. Review Comments to the Author

Reviewer #1: The paper is well-written and covers a relevant topic. The minor adjustment I suggest is related to the Strenghts and Limitations section, rows 588-590: "While the online survey limited the in depth probing characteristic of interview studies, it proved an efficient and appropriate method for gathering perceptions from a large, geographically dispersed, and diverse international team, aligning with our aim to broadly evaluate the IKT process (25).". Since only four respondents lived in LMIC's, there is indeed a limitation in stating that the method is appropriate from gathering perceptions from geographical disperse and diverse teams (i.e. 5 out of 9 members from South America and Africa didn't respond and we don't understand exaclty why). That being said, while the survey seems a promising method, you could explore whether translation of the survey or other inclusivity methods would allow for broader Global South participation.

Finally, while the authors didn't provide access to full data, they justified by citing confidentiality purposes. Given that the team is not big, access to full data could foster identification of respondents and all data from the survey was summarized so I believe the justification provided is enough.

Reviewer #2: 1. The paper presents an evaluation of the IKT methods for a particular project based on a survey of members of the IKT team. The survey and descriptive statistics provided are sound and overall, I am not sure they fully support the conclusions. The survey is of the perception of the team members of the impact of the IKT approach, and the process aspects that worked and didn’t work. But the actual use impact of the IKT approach is not assessed. For example, the participants report anticipates use of the guideline extension but there is no discussion if this was observed, however it seems to be reported as if this is occurring. It would be interesting to know if it is indeed occurring. But no doubt there are time constraints of the project, as it will take some time to know if people are using the guideline extension. Perhaps a more accurate representation of this evaluation is that it is a process evaluation. Then the findings and conclusions can focus on the implications of what worked or didn’t work for this type of IKT team. Which is present for sure, just could expand on that more. I also would have liked to see the evaluation be guided by some of the theories or frameworks in the literature, this might have helped with the more specific/accurate framing of the impact. For example, there is a community engagement assessment tool based on the REAP model that might have been useful to integrate into the analysis. (https://research.ucalgary.ca/toolkit) Or from the D&I literature, something like the RE-AIM might have helped to shape the assessment and analysis. Or perhaps another theory of change – if the team was using a theory of change they could present that here and what they know so far. The governance model presented early on might be interpreted this way? The findings could be considered in light of that perhaps?

2. Descriptive statistics are appropriate.

3. No concerns

4. The manuscript is clear and well written. There are a couple of typos here and there but overall, no concerns here. One overall comment is to make it clearer what you are (and not) doing with this paper. In line 218/19 you state the purpose of the paper. But I don’t know that is totally accurate to what you report. Do you really assess how the IKT approach reflected JEDI? My take away was more around (similar to comment 1) if the IKT approach used in the project was effective? What you learned about how to use it more effectively to be JEDI. And given the IKT focus, I wonder if the balance of background should lean a little more to IKT?

6. PLOS authors have the option to publish the peer review history of their article (what does this mean? ). If published, this will include your full peer review and any attached files.

**Do you want your identity to be public for this peer review?** For information about this choice, including consent withdrawal, please see our Privacy Policy .

Reviewer #1: No

Reviewer #2: No

Figure Resubmissions:

---

## [Decision Letter · Decision Letter 1]

6 Jan 2026

Evaluating the perceived outcome and impact of an integrated knowledge translation approach in the development of an equity reporting guideline: a cross-sectional survey

PGPH-D-25-02783R1

Dear Mr Dewidar,

We are pleased to inform you that your manuscript 'Evaluating the perceived outcome and impact of an integrated knowledge translation approach in the development of an equity reporting guideline: a cross-sectional survey' has been provisionally accepted for publication in PLOS Global Public Health.

Best regards,

Miqdad Asaria, Ph.D

Academic Editor

Thank you for thoroughly addressing all reviewer comments. I am delighted to accept this valuable contribution for publication. Feel free to address the minor typos highighted by reviewer 2

Reviewer Comments (if any, and for reference):

Reviewer's Responses to Questions

**Comments to the Author**

1. If the authors have adequately addressed your comments raised in a previous round of review and you feel that this manuscript is now acceptable for publication, you may indicate that here to bypass the “Comments to the Author” section, enter your conflict of interest statement in the “Confidential to Editor” section, and submit your "Accept" recommendation.

Reviewer #1: All comments have been addressed

Reviewer #2: (No Response)

2. Does this manuscript meet PLOS Global Public Health’s publication criteria ? Is the manuscript technically sound, and do the data support the conclusions? The manuscript must describe methodologically and ethically rigorous research with conclusions that are appropriately drawn based on the data presented.

Reviewer #1: Yes

Reviewer #2: Yes

3. Has the statistical analysis been performed appropriately and rigorously?

Reviewer #1: Yes

Reviewer #2: Yes

4. Have the authors made all data underlying the findings in their manuscript fully available (please refer to the Data Availability Statement at the start of the manuscript PDF file)?

Reviewer #1: No

Reviewer #2: Yes

5. Is the manuscript presented in an intelligible fashion and written in standard English?

Reviewer #1: Yes

Reviewer #2: Yes

6. Review Comments to the Author

Reviewer #1: The authors observed all recommendations and justified not being able to disclose all data.

Reviewer #2: Thank you for addressing my comments and feedback. The only comment you did not address was that the introduction is a little heavy on health equity and light on iKT and JEDI enhancement to it - evidence for the need. Often folks say IKT is inherently JEDI, clearly that is not true but maybe talk about that a bit based on the IKT literature.

The introduction could use a little editing for flow, and to bring the reader more quickly into the point of the paper. Some of the other phases of the research could be explained in that section too. Leaving the study design to focus on the design of this specific phase of the study that is being reported on in this paper. Which I believe is phase 2 but that is not totally clear.

A few other line edits/feedback to consider

132 this sentence: We conducted a cross-sectional survey of STROBE-Equity project members (n=68) – implied to me when I read the abstract as the n of the survey, quick fix, use a large N here.

241 The section with the heading Survey Development is more than just about the development of it, maybe Survey Description or something would be more accurate

263 Data collection heading does not need methods.

272 The last sentence would make more sense ending the previous section and take out the word described.

274 I would take out heading (Survey administration) and just have this paragraph under data collection

280 – 310 seems more like background then of the methods relevant to this paper (see overall comment above).

312 delete the extra our “…to guide our the evaluation of our IKT…”

325 Maybe this section could be in the background before methods also, it is not specifically relevant to the survey project being reported (alternatively this and the governance section maybe could be reported first under methods as set up).

7. PLOS authors have the option to publish the peer review history of their article (what does this mean? ). If published, this will include your full peer review and any attached files.

**Do you want your identity to be public for this peer review?** For information about this choice, including consent withdrawal, please see our Privacy Policy .

Reviewer #1: No

Reviewer #2: No
